

# Late Jurassic teeth of plesiosauroid origin from the Owadów-Brzezinki Lägerstatte, Central Poland

Łukasz Weryński[1] and Błazej Błazejowski[2]

[1] Doctoral School of Exact and Natural Sciences, Insitute of Geological Sciences, Jagiellonian University Cracow, Kraków, Lesser Poland Voivodeship, Poland
[2] Institute of Paleobiology, Polish Academy of Sciences, Warszawa, Masovia Voivodeship, Poland

## ABSTRACT

Owadów-Brzezinki is currently one of the most promising Upper Jurassic sites in Central Poland, with a wide array of both vertebrate and invertebrate fossil fauna present. The discoveries of large-bodied marine reptiles fossils such as ichthyosaurs, turtles, and marine crocodylomorphs attracted attention to the location. A particular Mesozoic marine group, plesiosaurs, remained to be found, and in this report, we note four isolated teeth with distinguishing apicobasal ridging pattern and elongated, conical shape characteristic for plesiosaurians. The outcomes of the Principal Coordinates Analysis (PCoA) of the largest and most complete tooth specimen ZPAL R.11/OB/T4 enabled us to confirm its classification as Plesiosauroidea. This discovery affirms the importance of the site as the area of mixing between Boreal and Tethyan faunas, expanding the broad spectrum of fossil taxa found in this location. Together with previous findings of plesiosaur material in a nearby region, it provides the evidence for the presence of Plesiosauroidea in Owadów-Brzezinki Lägerstatte.

# INTRODUCTION

Plesiosaurians are an iconic order of marine predators with compact four-flippered bodies and necks of varying lengths. They are known in the fossil record from the Upper Triassic up until the Upper Cretaceous (*Williston, 1914*; *Benson et al., 2013b*; *Wintrich et al., 2017*). In the Jurassic of Europe, especially the Upper Jurassic, many findings of taxa belonging to this group are well documented. Plesiosaurs have a good fossil record in Europe (*O'Keefe, 2004*; *Vincent, Bardet & Morel, 2007*; *Benson & Bowdler, 2014*; *Foffa, Young & Brusatte, 2018*; *Sachs, Klug & Kear, 2019*), also with a high prominence in the arctic Svalbard (*Knutsen, Druckenmiller & Hurum, 2012*; *Benson et al., 2013a*; *Roberts et al., 2017*, *2020*). However, this group is rather poorly represented and documented in Poland (*Madzia, Szczygielski & Wolniewicz, 2021* and references therein). To date, plesiosaur remains from Poland entail isolated teeth from the Aalenian of Wolin (*Deecke, 1907*), Bathonian of Jastrząb (*Rehbinder, 1913*), Oxfordian of the Zalas Quarry (*Molenda, 1997*; *Borszcz & Zatoń, 2009*; *Lomax, 2015*; *Bardet, Fischer & Machalski, 2015*), and Wapiennik (*Groß, 1944*; *Tyborowski, 2019*). Vertebrae are represented from the Callovian of Brzostówka (*Hirszberg, 1924*) and Kimmeridgian of Piekło (*Pusch, 1837*; *Hirszberg, 1924*). There is a

Corresponding author
Łukasz Weryński,
lukaszwerynski@doctoral.uj.edu.pl

report of a partial cranium from the Oxfordian of Częstochowa (*Maryańska, 1972*; *Tyborowski, 2019*). Also, unspecified remains have been reported from the Oxfordian of Inowrocław (*Jentzsch, 1884*), along with the various unpublished remains from the (?) Bathonian of Faustianka, Bathonian of Ogrodzieniec, Callovian of Bolęcin, Oxfordian of Mirów, and undetermined Jurassic strata of Młynka (*Madzia, Szczygielski & Wolniewicz, 2021*). Recently, a pectoral vertebra from the Kimmeridgian site of Krzyżanowice (*Tyborowski & Błażejowski, 2019*) was ascribed to Plesiosauria with possible placement in the family Cryptoclididae (*Madzia, Szczygielski & Wolniewicz, 2021*).

The Owadów-Brzezinki quarry has recently become a notable and promising fossil-bearing site (*Błażejowski et al., 2020*), with numerous finds of Late Jurassic vertebrate animals (*Tyborowski, Błażejowski & Krystek, 2016*). Marine reptiles in this faunal assemblage are represented by ichthyosaurs, metriorhynchid crocodylomorphs (*Tyborowski, Błażejowski & Krystek, 2016*), and pancryptodiran turtles (*Szczygielski, Tyborowski & Błażejowski, 2018*). However, no plesiosaur remains have been identified at this location so far. The palaeontological site of the Owadów-Brzezinki has been referred to as a new "taphonomic window" for the Upper Jurassic, providing insights into the distribution of Late Jurassic vertebrates in the Central European archipelago.

In this brief article we report the presence of Plesiosauria, based on the discovery of four isolated plesiosaur teeth, displaying characteristic features for this clade.

## GEOLOGICAL SETTING

The Owadów-Brzezinki quarry (51°22′27″N, 20°8′11″E) is an active open-pit marl and limestone mine, located in central Poland in the Łódzkie Voivodeship (Opoczno County) in the NW margin of the Holy Cross Mountains (Fig. 1). This palaeontological site is one of the most important recent palaeontological discoveries from Poland (*Kin et al., 2013*; *Błażejowski, Gieszcz & Tyborowski, 2016*). Remarkably well-preserved fossils of marine and terrestrial organisms of Late Jurassic (Tithonian) age, many of them new to science, provide a good opportunity to study the taphonomy of the ecosystem, evolution and migration of taxa, and paleoenvironmental changes (cf. *Błażejowski, Gieszcz & Tyborowski, 2016*; *Błażejowski et al., 2019*; *Wierzbowski et al., 2016*). Especially interesting is the fact that new species, endemic to this site, are constantly being discovered, such as the lobster-like decapod crustaceans (*Feldmann, Schweitzer & Błażejowski, 2015*; *Błażejowski, Gieszcz & Tyborowski, 2016*) and xiphosuran arthropods (*Kin & Błażejowski, 2014*; *Błażejowski, 2015*; *Błażejowski et al., 2019*, *2020*), constituting one of the largest accumulation of Jurassic horseshoe crabs discovered so far. Most prominent taxa of vertebrates discovered so far are represented by the ichthyosaur *Cryopterygius kielanae* (*Tyborowski, 2016*; *Undorosaurus* according to *Zverkov & Efimov, 2019*, and *Zverkov & Jacobs, 2021*) and the pancryptodiran turtle *Owadowia borsukbialynickae* (*Szczygielski, Tyborowski & Błażejowski, 2018*). Other vertebrate taxa are represented by Actinopterygii and Elasmobranchii (*Kin et al., 2013*; *Błażejowski et al., 2015*) and marine crocodylomorphs (*Błażejowski, Gieszcz & Tyborowski, 2016*), with additional shore fauna represented by insects, terrestrial crocodylomorphs, and possibly pterosaurs (*Kin et al., 2013*). The Owadów-Brzezinki section is located within both the Brzostówka marls of the

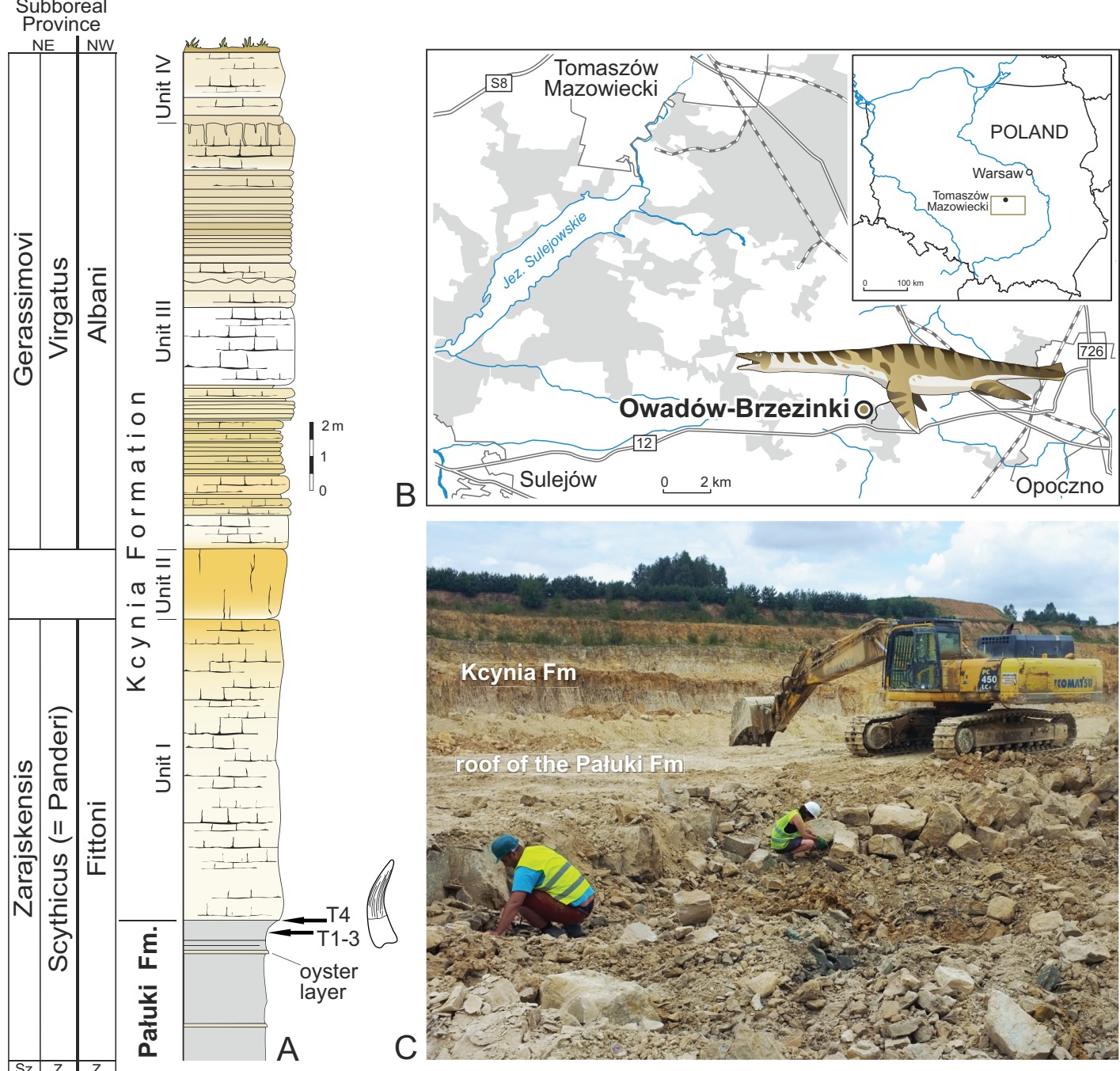

**Figure 1 Location of the site and lithological succession in the study area with tooth bearing interval marked.** (A) Lithological succession and biostratigraphy of the Owadów-Brzezinki Quarry. The topmost part of the Pałuki Fm and overlying limestones of the Kcynia Fm (Units I–IV). (B) Road map with the location of the Owadów-Brzezinki site and its proximity to Tomaszów Mazowiecki in Central Poland. (C) General view of the Owadów-Brzezinki section (paleontological field work in the uppermost part of the Pałuki Fm). Abbreviations: Fm, Formation; Sz., Subzone; Z., Zone.

topmost part of the Pałuki Formation (Fm) and the overlying limestones of the Kcynia Fm (*Błażejowski, Gieszcz & Tyborowski, 2016*). The uppermost part of the Pałuki Fm and the overlying limestones of the Kcynia Fm, including the Sławno Limestone Member (Mb), "*Corbulomima* limestones", and a horizon of "serpulid" beds, exposed in the section (*Kutek, 1994*; *Matyja & Wierzbowski, 2016*). The sedimentary pattern observed in the Owadów-Brzezinki section indicates a gradual marine regression revealed by a transition from offshore to coastal and lagoonal settings but its uppermost part was deposited during a short-term marine transgression and the re-appearance of coastal environments (*Błażejowski, Gieszcz & Tyborowski, 2016*; *Wierzbowski et al., 2016*).

The uppermost part of the Brzostówka Marl Mb of the Pałuki Fm from the Owadów-Brzezinki quarry (ca. 4 m thick) consists of black, blue-greyish and yellow-bluish marls with the intercalation of thin oyster-bearing and marly limestone beds (*Błażejowski, Gieszcz & Tyborowski, 2016*; *Wierzbowski et al., 2016*). The marls yielded abundant marine microfossils, bivalves, ammonites, decapod crustaceans and fish (*Błażejowski, Gieszcz & Tyborowski, 2016*). The overlying limestones of the Kcynia Fm have been subdivided into four lithological units (cf. *Błażejowski, Gieszcz & Tyborowski, 2016*).

According to the stratigraphical studies of *Kutek (1994)* and *Matyja & Wierzbowski (2016)* based on the ammonite fauna, the lower part of the Owadów-Brzezinki deposits is dated to the regularis horizon (the uppermost part of the Brzostówka Marl Mb of the Pałuki Fm) and zarajskensis horizon (unit I of the Sławno Limestone Mb of the lowermost part of the Kcynia Fm) of the Zarajskensis Subzone of the Scythicus (Panderi) Zone of the Middle Volgian, as well as to the Fittoni Zone from the "Bolonian" zonation of England (*Matyja & Wierzbowski, 2016*). The upper part of the Owadów-Brzezinki section (units III and IV belonging to the "*Corbulomima* limestones" and "serpulid" beds, respectively) has, in turn, been assigned to both the Gerassimovi Subzone of the Virgatus Zone of the Middle Volgian and the Albani Zone of the "Portlandian" (*Matyja & Wierzbowski, 2016*). Owadów-Brzezinki has recently attracted much attention not only due to the exquisite quality and quantity of preserved fossils, but also due to its palaeogeographic significance —this site is proposed to encompass an important area, located on the border of the Boreal/Subboreal and Tethyan realms, where the mixing of temperate and tropical faunal biota occurred (*Błażejowski, Gieszcz & Tyborowski, 2016*; *Błażejowski et al., 2023*; *Matyja & Wierzbowski, 2016*).

## METHODS AND TERMINOLOGY

All teeth have been prepared manually. Specimens were coated with sublimed ammonium chloride to accentuate the fine structure of the enamel, and then photographed using a Nikon D5 (55 mm f/2.8) digital camera (Figs. 2A–2O). The collected material is housed at the Institute of Paleobiology, Polish Academy of Sciences in Warsaw (ZPAL R.11). The terminology of tooth orientation is based on marine reptiles teeth studies (*Zverkov et al., 2018*; *Madzia, 2020*) with the following terms used: apical—towards the apex of the tooth crown; basal—towards the tooth crown base; mid-crown—approximately centrally between the crown apex and base; mesial—anteriorly/anteromedially along the tooth row, towards the tip of the animal's mouth; distal—posteriorly/posterolaterally along the tooth

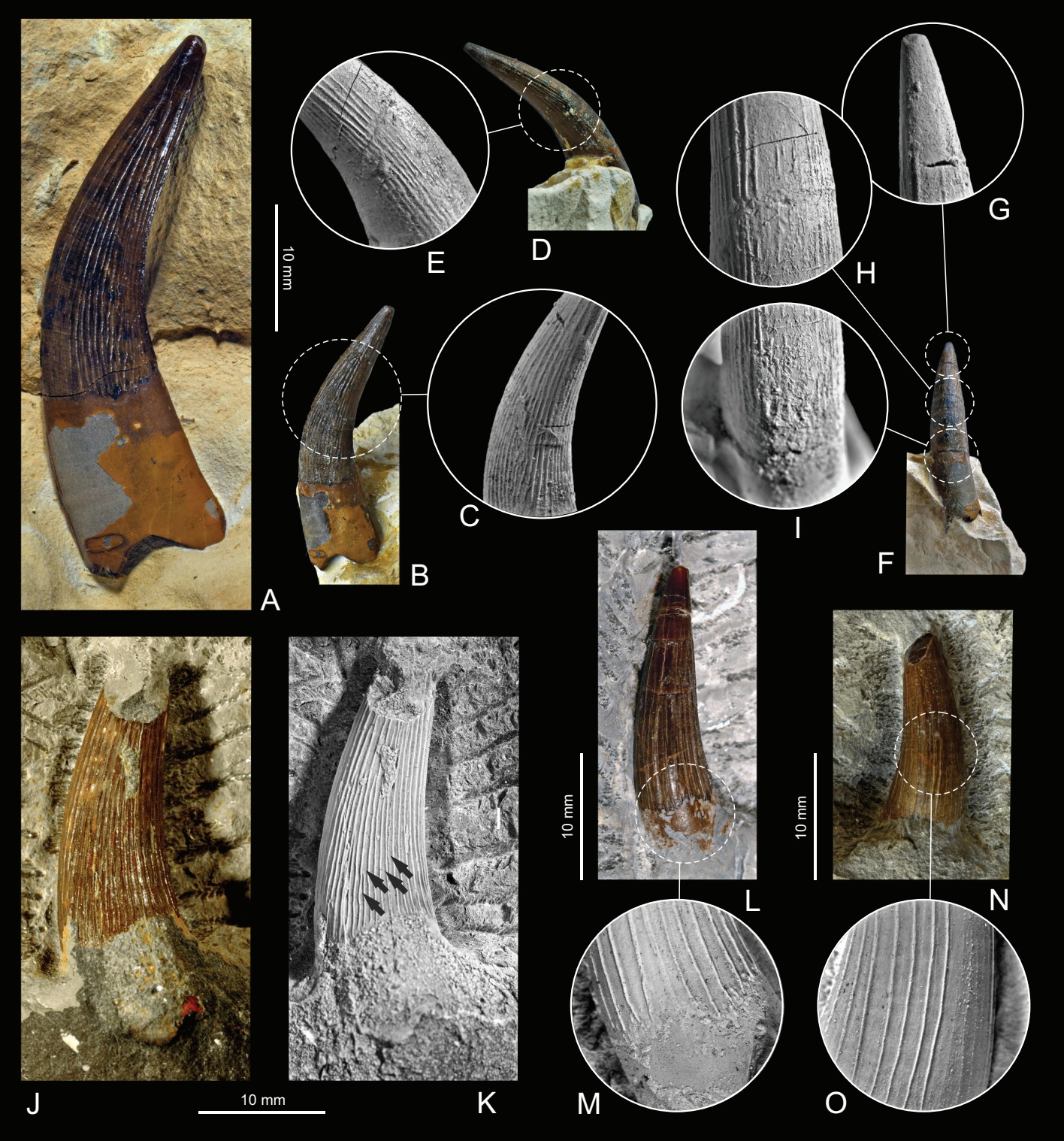

**Figure 2 Teeth specimens featured in this study.** (A–F) ZPAL R.11/OB/T4, (A) overview; (B) distal face and apicobasal ridges in close view (C). (D) Labial face and apicobasal ridges in close view (E). (F) Mesial face and close view of apical (G), mid-crown (H) and basal (I) teeth sections. (J) ZPAL R.11/OB/T3 in distal view with prominent, sharp apicobasal ridges highlighted (K). (L) ZPAL R.11/OB/T2 in labial view with close view of teeth base (M). (N) ZPAL R.11/OB/T1 in lingual view with close view of mid crown apicobasal ridges (O).

row, towards the jaw articulation; labial—towards the animal's exterior of the mouth; lingual—towards the tongue.

Principal coordinate analysis (PCoA) was used to explore the morphospace occupied by the specimen ZPAL R. 11/OB/T4. This sample was the largest and with the most complete crown, hence it was deemed the only specimen viable for such an analysis. The PCoA enables exploration of multiple morphological characteristics of the teeth at once. In this study, we used the modified data matrix of *Madzia, Szczygielski & Wolniewicz (2021)*, which was originally published by *Foffa et al. (2018)*. The PCoA was conducted using PAST 4.10 software (*Hammer, Harper & Ryan, 2001*). The descriptions of the continuous and discrete characters, which are used in this data matrix, can be found in the supplement of *Foffa et al. (2018)*. The continuous characters used in this analysis have been z-transformed. Gower similarity index (*Gower, 1971*) was employed, since it is well suited for data with both continuous and discrete variables, as noted by *Madzia, Szczygielski & Wolniewicz (2021)*. The modified dataset with included ZPAL R. 11/OB/T4 data is available in the Supplemental File (SM_OW), and raw PCoA results, containing eigenvalues and diagrams, can be found in Supplement 2.

## SYSTEMATIC PALAEONTOLOGY

**SAUROPTERYGIA** *Owen, 1860*

**PLESIOSAURIA** *de Blainville, 1835*

**PLESIOSAUROIDEA** *Gray, 1825*

**CRYPTOCLIDIDAE?** *Williston, 1925*

**Material**: four isolated teeth, labelled ZPAL R. 11/OB/T1–ZPAL R. 11/OB/T4

**Locality and horizon**: Owadów-Brzezinki (Central Poland), Tithonian, upper Pałuki Fm and lower part of the Unit 1 of the Kcynia Fm.

Specimens were collected during various fossil excavations. ZPAL R. 11/OB/T1, ZPAL R. 11/OB/T2, and ZPAL R. 11/OB/T3 were collected from dark grey marls belonging to the Pałuki Fm, while the best preserved specimen (ZPAL R. 11/OB/T4) was found in the limestone at the base of the Kcynia Fm, within a few centimetres above the boundary with the Pałuki Fm, which consists of chalky limestones representing the Unit I. Due to the collection of the specimens at different times and strata, connected with the exploitation of the quarry, it is likely that ZPAL R. 11/OB/T1–ZPAL R. 11/OB/T3 and certainly ZPAL R. 11/OB/T4 belonged to different individuals. The specimens are partially encased in the matrix, which was not removed to avoid damaging specimens, which at the time limits the observable characteristics of the specimens. The enamel of the teeth and overall morphology are generally excellently preserved, even though all the specimens are incomplete. While apicobasal ridges are very prominent, with sharp, prominent edges the enamel appears mostly smooth, with no additional striae present. Ridging is visible on the mesial, labial, lingual and distal faces of the preserved fragments. All teeth can be described as conical in overall morphology, with oval-to-subcircular cross section. Independent of the host rock color, all reported specimens appear to have the same dark brown coloration. This is easily explainable in the case of specimens ZPAL R. 11/OB/T1, ZPAL R. 11/OB/T2 and ZPAL R. 11/OB/T3, as they were acquired from similar strata of the same formation, but in the case of specimen ZPAL R. 11/OB/T4, coming from the Unit I of the Kcynia Fm,

it has more interesting implications, and it appears that the fossilization process led to a similar outcome in both cases—it is explainable by fact that the Unit I has been described by *Wierzbowski et al. (2016)* as deposited in a standard marine setting, transitional to nearshore environments of the upper units, so the conditions were similar. This form of preservation is also typical for other teeth from this interval.

**Specimen description**

**ZPAL R. 11/OB/T4:** The largest and best-preserved specimen (Figs. 2A–2I), characterized by a complete crown, measures 47 mm in total length (including preserved part of the root), with the apicobasal crown height of 28.6 mm. The base of the tooth can be measured at 10.23 mm in mesiodistal diameter (MD), 8 mm in labiolingual length (LD), with mid-crown length of 7 mm. The apicobasal length/basal diameter (MD+LD/2)—crown ratio (CR) is 3.13. The overall shape of the crown can be described as conical, elongated with slight lateral compression, leading to ovaloid cross section. It is the only specimen that has a part of the root preserved. This specimen is characterized by a very sharply defined enamel boundary between the tooth crown and root. The root is proportionally narrow mesiodisally when compared to the crown, only slightly wider than the tooth base, measuring 13 mm in mesiodistal diameter. Completely exposed sections consist of the lingual, mesial, and distal faces, with mostly exposed apical part, allowing observation of the slightly ovaloid cross-section. The apex appears strongly recurved, and the overall curvature (convex/concave length ratio) of the tooth is 1.21. Apicobasal ridges are finer than the coarser ridges in ZPAL R. 11/OB/T3, but still very prominent, and they appear rather irregular in pattern, with some present along the whole crown height, and some only in certain segments. The ridging appears more prominent on the lingual rather than labial face of the crown. In the basal section of the crown, the ridging appears to ascend apically at a slight angle, especially in the lingual plane, while the apical part is devoid of ridging, which disappears 2 mm from the apex. There is a shallow but wide apicobasally directed groove present on the lingual surface of the root, which appears rather regular in shape, and we consider it to be an integral characteristic of root.

**ZPAL R. 11/OB/T3:** This specimen (Figs. 2J and 2K), measuring 24 mm in total length, consists of mostly intact crown with a part of the apex and base missing. Measuring 10 mm in mesiodistal diameter at mid-section and 12 mm in mesiodistal diameter near the base, the specimen appears especially robust, with a nearly sub-circular cross section. Labial, distal, and apical faces with a part of the lingual face are exposed. It has well-preserved apicobasal ridges, which appear especially prominent and dense. This prominence can be attributed to the excellent preservation of enamel. The ridges, in contrast to smaller teeth, appear to be slightly irregular in their form, with a winding, intertwining structure. The mesiodistal curvature can be characterized as lesser in comparison to ZPAL R. 11/OB/T4.

**ZPAL R. 11/OB/T2:** Measuring 23 mm in total length, this specimen (Figs. 2L and 2M) has almost the complete crown preserved, with the root missing. The shape can be described as elongated and conical, with a minor labiolingual compression. This specimen has its labial and mesial faces exposed. Its mid-crown diameter in the mesiodistal plane is 6 mm, with a MD of 8 mm and LD of 6.3 mm. The CR is 3.21. Apicobasal ridges appear

straight and regularly developed, while the mesiodistal curvature is only slight. The crown appears to possess a slightly worn apex.

**ZPAL R. 11/OB/T1:** The smallest of the featured specimens and with the poorest preservation (Figs. 2N and 2O), measures 16 mm in total length with the mesiodistal mid-crown diameter of 6 mm. The preserved part of the crown can be characterized as conical and slightly ovaloid in cross section. Only the upper portion of the tooth crown is preserved, and the apex of the tooth is missing. Nevertheless, even if fully preserved, this specimen would likely be smaller than ZPAL R. 11/OB/T2, ZPAL R. 11/OB/T2 and ZPAL R. 11/OB/T4. The missing apical section in this case can likely be a result of tooth wear. Exposed are the labial, mesial, and distal faces. Tooth curvature in the mesiodistal plane can be characterized as slight. Apicobasal ridges appear especially prominent on the distal surface, and they can be described as less densely packed than in larger specimens (*e.g.*, ZPAL R. 11/OB/T3).

### Principal coordinate analysis

Tooth morphology within various groups of marine reptiles is often distinctive. Therefore, plotting isolated tooth specimens in a morphospace may allow their attribution to particular taxa. We have been able to score 19 out of the possible original 20 characters used by *Madzia, Szczygielski & Wolniewicz (2021)* for specimen ZPAL R. 11/OB/T4. The morphospace occupation plot (Fig. 3) depicts the occupation of the morphospace of ZPAL R. 11/OB/T4 in relation to other plesiosaurian taxa. Comparisons with inferred closely related taxa (based on observed morphology) enable a detailed localization of the specimen in the morphospace. The negative side of the x axis is occupied by the Pliosauridae, while the positive side covers the morphospace occupied by the Plesiosauroidea. ZPAL R. 11/OB/T4 is recovered within the Plesiosauroidea morphospace.

## DISCUSSION

Mesozoic marine reptile teeth exhibit a wide array of morphologies (*Massare, 1987*), which correspond to their varied ecological niches. The shape of the discovered teeth (Fig. 4) is generally elongated, slightly recurved with a sub-circular cross-section. Roots are either missing (ZPAL R. 11/OB/T1–ZPAL R. 11/OB/T3) or poorly preserved (ZPAL R. 11/OB/T4), but the preserved portion of the root of ZPAL R. 11/OB/T4 indicates narrow, elongated roots, barely wider than the crown. This characteristic, linked to the relative elongation of the teeth, may suggest adaptations to a piscivorous diet (*Massare, 1987, 1997*). Intriguingly, the teeth exhibit lateral compression, with a ML/LL ratio of 0.78 for ZPAL R. 11/OB/T4 and 0.79 for ZPAL R. 11/OB/T2. Considering the relative proportions of the crowns and visible wear of apices, the described teeth can be long to Pierce II/ generalist guild of *Massare (1987, 1997)*, which characterizes marine reptiles preying on small prey. According to the morphospace analysis conducted by *Fischer et al. (2022)*, predators possessing complicated tooth structures, such as the richly ornamented teeth observed in the specimens form Owadów-Brzezinki, typically specialize in small prey. In contrast, predators of large prey tend to have teeth with a simpler morphology (the exception to this rule appear to be the Pliosauridae).

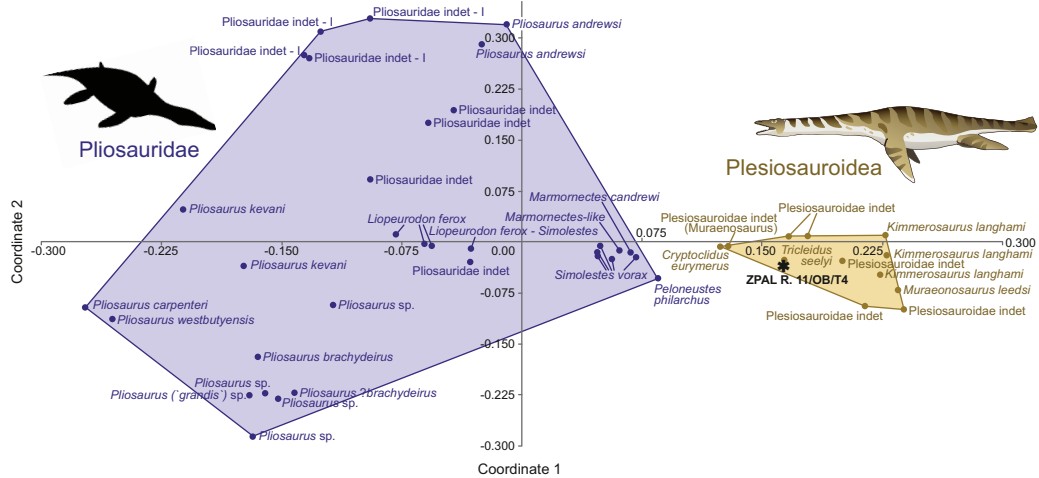

**Figure 3** ZPAL R.11/OB/T4 (black star) morphospace occupation among marine reptiles of Jurassic, visualization of results of PCoA, segregation along principal coordinates 1 and 2. Based on the modified dataset from *Madzia, Szczygielski & Wolniewicz (2021)*, originally from *Foffa et al. (2018)* compared against the Plesiosauria. The silhouette of the Pliosauridae taken from Phylopic, by Nobu Tamura, vectorized by T. Michael Keesey (CC BY 3.0 SA), link: https://www.phylopic.org/images/91036b62-ca14-4c61-9cf6-67c4940d8c02/peloneustes-philarchus.

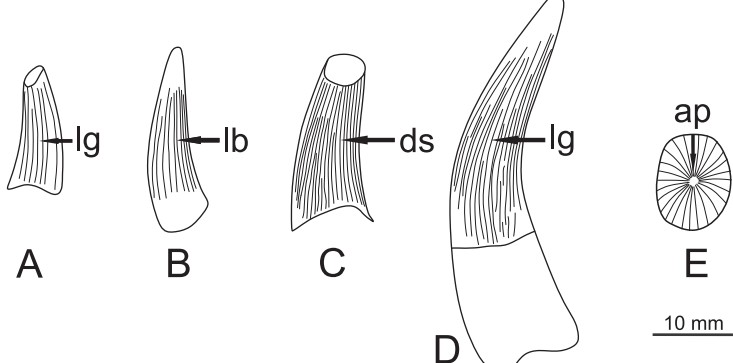

**Figure 4** Specimen teeth sketch figure showing general morphologies, with each face highlighted. (A) ZPAL R.11/OB/T1, lingual view. (B) ZPAL R. 11/OB/T2, labial view. (C) ZPAL R. 11/OB/T3, distal view. (D) ZPAL R. 11/OB/T4, lingual view (E) ZPAL R. 11/OB/T4, apical view. Abbreviations: lg, lingual; lb, labial; ds, distal; ap, apical.

One of the defining characteristics of the Plesiosauria, in contrast to the Ichthyosauria and Crocodylomorpha, is the presence of very prominent, elevated apicobasal ridges which in some cases are also accompanied by shearing carinae (*Massare, 1987*; *Lomax, 2015*). Long-necked Plesiosauroidea, are characterized by their elongated and conical teeth with pointy apices (*Massare, 1987*, *1997*). The examined teeth exhibit a crown height to basal greater than three which is another characteristic for plesiosaurians, noted by *Massare (1987)*. An especially striking feature is the presence of strong apicobasal ridges, which are visible along the length of the entire tooth crown, except for the apex region. Additionally, the carinae-like ridges merge with the apicobasal ridges in the mesial and distal regions,

which can be detected through hand examination. Furthermore, the apicobasal ridges are narrow and sharp, as is typical of the Plesiosauria (*Massare, 1987*). The crown ratio (CR) is 3.21 for ZPAL R. 11/OB/T2 and 3.13 for ZPAL R. 11/OB/T4, which lies within the spectrum of ratios described by *Massare (1987)* as characteristic for plesiosaurian teeth.

One of the most peculiar characteristics of many aquatic predator teeth is the presence of apicobasal ridges (*McCurry et al., 2019*), which develop longitudinally in various forms and are sometimes accompanied by winding striae on the surface of the teeth. These structures have yet to be understood in depth, although it is suggested that they enhance the mechanical properties of teeth in an aquatic setting (*Ciampaglio, Wray & Corliss, 2005*; *McCurry et al., 2019*). The apicobasal ridges are often used as a distinguishing feature (*Brown, 1981*), which, in addition to overall morphology, allows for a rough differentiation of genera, particularly when only teeth are available. In the examined samples, the apicobasal ridges appear to form along almost the entire apicobasal length of the tooth crown (except for the very apex) and are present mesially, lingually, labially, and distally, encompassing the entire circumference of the teeth. Excluding the ridges, the crown surfaces are smooth, with no visible striations. In the studied specimens, the ridges seem continuously more prominent in larger teeth, and the general structure becomes more winding and complex in larger specimens. This can be possibly ascribed either to: (1) tooth allometry, as a larger size requires more structural support to handle the stress induced by struggling prey (*McCurry et al., 2019*); (2) ontogenic variation, with ridges becoming more prominent in larger, older specimens, or (3) some form of heterodonty, in which larger teeth of a single individual exhibit more prominent ridges than smaller teeth. A similar relationship can be observed in tooth curvature, which becomes more prominent as the studied tooth specimens get larger. However, this may be an effect of preservation, as smaller teeth are less intact, especially compared to ZPAL R.11/OB/T4, hence assessing the degree of curvature *in vivo* might be challenging, with additional possibility of diagenetic processes influencing the shape and pronouncing the curvature in larger specimens.

## Palaeobiogeographic context and morphological characteristics in comparison to other plesiosaurian dental material

Based on the paleobiogeographic context, the teeth described herein most likely belonged to the family Cryptoclididae within the Plesiosauroidea. Cryptoclididae are a family of plesiosaurs common in the Middle to Late Jurassic (*Roberts et al., 2020*), and are well-represented in the fossil record, including findings from England and Svalbard (*Knutsen, Druckenmiller & Hurum, 2012*; *Benson & Bowdler, 2014*; *Foffa, Young & Brusatte, 2018*; *Roberts et al., 2017*, *2020*). They constituted one of the dominant groups of plesiosaurs in the Middle and Late Jurassic of the Northern Hemisphere until their decline at the Jurassic-Cretaceous transition (*Benson & Druckenmiller, 2013*). As such, in the palaeobiogeographical context, this family is likely to inhibit Owadów-Brzezinki, since large marine reptiles of the Mesozoic margin of Holy Cross Mountains are considered to have been mainly of the Boreal realm provenance (*Tyborowski, 2016*). Therefore, one of the closest analogues to the Owadów-Brzezinki site in the context of plesiosaur species identification can be the Upper Jurassic Kimmeridge Clay. This location also encompasses

the Tithonian, belongs to the Boreal province, and bears various marine reptile taxa (*Benson & Bowdler, 2014*; *Foffa, Young & Brusatte, 2018*), often plesiosaurs. It should be noted that the Kimmeridge Clay represents a much larger time interval than the site tacked in this study. The Kimmeridge Clay and formations of a similar age in the United Kingdom yield remains of two families of plesiosaurs: the long-necked Cryptoclididae and the large-headed macro predatory Pliosauridae (*Foffa, Young & Brusatte, 2018*). The elongated form of the teeth examined here closely corresponds to the long-necked plesiosaur teeth from the Coralline Gap Formation (which underlies the Kimmeridge Clay) that were described by *Foffa, Young & Brusatte (2018)*, and belong to the Cryptoclididae. The Cryptoclididae have also been identified in the Slottsmøya Member of the Agardhfjellet Formation (*Roberts et al., 2017*) in Svalbard, which is of Tithonian-Berrasian age, also belonging to the Boreal province.

It is postulated that there are at least four genera of plesiosaurs *sensu stricto* (long-necked) present in the Kimmeridge Clay (*Benson & Bowdler, 2014*) that ought to be included in the family Cryptoclididae. The latest work by *Foffa, Young & Brusatte (2018)* postulates that the Cryptoclididae from the Coralline Group, which sits below the Kimmeridge Clay, are represented by *Muraenosaurus leedsi, Tricleidus seeleyi, Cryptoclidus eurymerus* and *Kimmerosaurus langhami*. In contrast to *Cryptoclidus* which exhibits reduced ridging, and *Kimmerosaurus*, characterized by almost completely reduced ridging, the teeth examined here are characterized by ridging present throughout the labial, distal, mesial, and lingual surfaces, excluding only the apex region. The observed ridging pattern, is much more similar to that seen in *Muraenosaurus* or *Tricleidus*, because those taxa exhibit complex and complete ridging on all faces of their teeth. *Mureanosaurus* dentition is much more elongated and conical (*e.g.*, *Foffa, Young & Brusatte, 2018*; *Foffa et al., 2018*: Supplement) than that of the specimens studied herein. *Tricleidus* exhibits a similar apicobasal ridging pattern as the specimens under investigation, with the reduction of ridging on the apical portion of mesial face and prominent ridging on other surfaces of the crown (*Foffa, Young & Brusatte, 2018*). However, the teeth of *Tricleidus* differ from the studied specimens as they are circular in cross section, and much more elongated with a higher CR, as noted in *Foffa, Young & Brusatte (2018)*.

There are members of the Cryptoclididae from Spitzbergen represented by dental material—*Spitrasaurus larseni* (see *Knutsen, Druckenmiller & Hurum, 2012*) and *Ophthalmothule cryostea* (see *Roberts et al., 2020*). The dental material of *S. larseni* exhibits very gracile crowns, D-shaped in cross-section (*Knutsen, Druckenmiller & Hurum, 2012*), quite dissimilar in overall morphology to the specimens presented here. The dental material of *O. cryostea* exhibits more similar proportions to the studied teeth (*Roberts et al., 2020*: Fig. 11), but the ridging pattern is comparatively fine and more prominent on the labial surface, while the examined specimens tend to show greater ridging on the lingual faces, instead.

*Colymbosaurus* is a genus well-represented in fossil record of England and Svalbard. There are three species of *Colymbosaurus*: *C. megadeirus* from England (*Seeley, 1869*; *Benson & Bowdler, 2014*), *C. svalbardensis* from Svalbard (*Persson, 1962*; *Knutsen, Druckenmiller & Hurum, 2012*; *Roberts et al., 2017*), and also C. sclerodirus (*Bogolubow,*

*1911*). Given its widespread presence, this genus appears to be a promising candidate. However, there is limited knowledge about the species as they are primarily recognized based on their postcranial skeleton, with only a single reported mandible fragment of C. *svalbardensis* (*Roberts et al., 2017*). Additionally, the lack of information on their dental morphology presents a significant challenge. The possibility of the material belonging to a plesiosaur family other than Cryptoclididae is also to be considered, as classification based on isolated teeth can bear a margin of error (*Knutsen, 2012*), and the data matrix provided by *Foffa et al. (2018)* lacks representation of some plesiosaur families, such as the Rhomaleosauridae. The option that the teeth described herein could belong to multiple taxa should also be considered.

## CONCLUSIONS

The results of the principal coordinate analysis (PCoA) and overall morphology provide strong evidence for the described teeth as belonging to a plesiosauroid. The most probable placement of this material is the Cryptoclididae, based on morphology and the paleobiogeographic context, but also other possible taxonomic affiliations cannot be excluded. In the light of this discovery, further exploration is necessary to uncover additional diagnostic material, particularly fossilized plesiosaur bones, which would solidify the identification. Finding fossil remains of plesiosaurians presents us with an exciting opportunity to expand our understanding of Owadów-Brzezinki. While this site has been compared to the Solnhofen Lagerstätte due to its exceptional preservation (*Kin et al., 2013*), the discovery of plesiosaurians sets it apart from its German counterpart. This discovery bears significant implications, as it confirms the presence of a large and diverse assemblage of macro vertebrate fauna, and reveals new avenues for research into the evolution and ecology of large predatory marine reptiles in Owadów-Brzezinki.

## ACKNOWLEDGEMENTS

We would like to thank Tomasz Szczygielski (Institute of Paleobiology, PAS) for many helpful suggestions during the early phase of this investigation. We express our thanks to Stanisław Kugler for finding specimen ZPAL R.11/OB/T4. Thanks are given to Aleksandra Hołda-Michalska (Institute of Paleobiology, PAS, Warszawa) for the improvement of the figures. We wish to acknowledge Daniel Madzia (Institute of Paleobiology, PAS) and Aubrey Roberts (University of Oslo) for their critical review and very helpful comments that improved the manuscript, and one anonymous reviewer. We express our thank to Mark Young, the journal Editor, for critical review and valuable suggestions.

### Funding

Travel and field work were supported by the Polish National Science Center Grant no. 2020/39/B/ST10/01489. The research in 2022 was supported by the Priority Research Area (WSPR.WGiG.1.5.2022.2) under the Strategic Programme Excellence Initiative at

Jagiellonian University. The funders had no role in study design, data collection and analysis, decision to publish, or preparation of the manuscript.

## Grant Disclosures
The following grant information was disclosed by the authors:
Polish National Science Center: 2020/39/B/ST10/01489.
Priority Research Area: WSPR.WGiG.1.5.2022.2.

## Competing Interests
The authors declare that they have no competing interests.

## Author Contributions
- Łukasz Weryński conceived and designed the experiments, performed the experiments, analyzed the data, prepared figures and/or tables, and approved the final draft.
- Błazej Błażejowski performed the experiments, prepared figures and/or tables, authored or reviewed drafts of the article, and approved the final draft.

## Data Availability
 The raw data is available in Fig. 2 and the Supplemental Files.

## Supplemental Information
Supplemental information for this article can be found online at http://dx.doi.org/10.7717/peerj.15628#supplemental-information.

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
