# Peer review of "Late Jurassic teeth of plesiosauroid origin from the Owadów-Brzezinki Lägerstatte, Central Poland"

_PeerJ, doi:10.7717/peerj.15628_

## Round 0.1 · original submission · Major Revisions

Dear authors,

Based on the opinions of the reviewers, I have made a decision of 'major revisions'. The reviewers have highlighted that more comparative studies are needed to make a confident identification. Reviewer three has also suggested a morphometric approach.

I look forward to receiving your revised manuscript.

·

Basic reporting

Thank you for the opportunity to review the manuscript by Łukasz Weryński and Błażej Błażejowski titled "Late Jurassic teeth of possible cryptoclidid origin from the Owadów-Brzezinki Lagerstätte, Central Poland". It is great to see a new plesiosaur material from the Upper Jurassic of Poland.

The language is mostly clear and the background was explained rather well. Overall, however, many parts of the text appear to be much longer than necessary and can be removed or substantially shortened before publication. In turn, there are parts that seem somewhat insufficient; for example, comparisons are rather superficial in my opinion.

Also, despite that the text is richly referenced, many references do not seem to be appropriately selected and somewhat leave the impression to have been selected rather arbitrarily. I highlighted one such example in the Introduction but similar parts can be spotted throughout the MS.

As for the structure of the article - the description was added to "MATERIAL, METHODS AND TERMINOLOGY", which is followed by "DISCUSSION" and then "CLASSIFICATION". I think that the descriptive part (preferably combined with detailed comparisons ideally involving a wide arrey of Late Jurassic taxa of Europe) should be included in the "CLASSIFICATION" section (which should be renamed "Systematic palaeontology". And then, the "Discussion" should follow.

Figures are clear and well-arranged, and all data necessary to assess the conclusions are provided.

Please, see the attached document for some comments to certain part.

Experimental design

The study certainly represents an original primary research. The research question is well-defined and relevant.

Validity of the findings

I have nothing to add with respect to the conclusions and agree that the teeth likely belong to cryptoclidids, though refering them to as Plesiosauroidea indet. might be more appropriate. I leave it up to the authors though.

Additional comments

I look very much forward to seeing this paper published in PeerJ. It fits perfectly within the scope of the paleontological section of the journal and I think that the study provides description of a very interesting and important material. However, major revisions are probably needed to make the article acceptable for publication.

Reviewer 2 ·

Basic reporting

Overall, the English grammar and sentence structure is good! However, there are certain areas where improvements need to be made. In the PDF, I have highlighted areas that need improvement and put in my suggestions for how to improve the writing. The references to the literature are mostly sufficient but see my comments for areas where I think there are missing references. The geologic background was excellent. The figures are well-done and illustrative of the important features. The structure of the paper is good, although I would move the identification section to before the discussion because it is part of the results.

Experimental design

With respect to the experimental design, there needs to be more comparative work done to confidently identify the isolated teeth as belonging to Cryptoclididae. The investigation could also be improved by demonstrating to readers of this paper that the isolated teeth indeed belong to Plesiosauria and not from other marine reptiles from the locality (although I agree that the teeth are plesiosaurian). For the methods, I would suggest you add how you did the comparisons- was it based on the literature, or did you compare the teeth at collections?

Validity of the findings

The conclusion is that these new teeth are cryptoclidid but I think more comparative work is in order, which you can add to your classification section. How are these new teeth different from microcleidid plesiosaurians? What about rhomaleosaurids? It is demonstrated that the teeth could not belong to Pliosaurus, but what about a small-bodied pliosauroid? Are there any characters that clearly demonstrate the teeth to be cryptoclidid? I am aware that microcleidids are known (thus far) only from the early Jurassic, but nobody knowns when they went extinct. It's plausible some may have persisted until the late Jurassic. I understand the biogeography argument, with cryptoclidid plesiosaurians being found in the UK, Norway, and Poland. But at the end of the day, the morphology is what is most important, and I advise to argue using morphological characters. Perhaps worst case, the teeth are ascribed to Plesiosauria but nothing more specific.

Additional comments

The manuscript is nicely written and structured in a way that makes sense. The biggest critique I have is just that the comparisons need to be more exhaustive and conclusive to allow for a family identification of the plesiosaur teeth.

Annotated reviews are not available for download in order to protect the identity of reviewers who chose to remain anonymous.

·

Basic reporting

Professional english is used throughout. There are few minor mistakes/rewording that should be picked up in the editorial process and I have highlighted a few suggestions - otherwise I have no issues here.

The paper lacks a few key cryptoclidid papers, for which I have provided the references for. These I believe are all open access so there should be no isssue in aquiring them. Some statements require supporting references - this is noted in the annotated PDF

The article structure needs some tweeking - I think it would be best - and shorten the paper if the descriptions are included in the systematic palaeontology section (which is currently in the discussion?). The figures are satisfactory - I have no issues here.

The paper provides evidene of the presence of cryptoclidids in the Late Jurassic of Poland and such is self-contained.

Experimental design

The paper fits the scope of the journal.

The paper defines a clear knowledge gap - which it fills in. However I have highlighted some points in the annotated PDF where this could be made more clear.

The figures and associated descriptions are ok, but more comparisons could be added. These have been noted in the annotated PDF. I believe the material is ethically sourced.

No analyses were completed in this paper - However it would be more interesting and lift up the paper even more - if the authors completed a principal components analysis using the data provided by: Foffa, D., Young, M.T., Stubbs, T.L., Dexter, K.G., and Brusatte, S.L. 2018c. The long-term ecology and evolution of marine reptiles in a Jurassic seaway. Nature Ecology & Evolution 2: 1548–1555.
I am not 100 % sure if it would be possible on the material as the teeth are incomplete - but might be worth investigating.

Validity of the findings

All underlying data are provided and observations are sound. Thes resulting conclusions are well stated.

Additional comments

Comments on the description:
- It is difficult to describe a specimen adequately without some form of comparison (example line 166). This happens throughout the description. Using words such as “narrow”, “robust”, “gracile” ect. Should only be used when comparing to something – otherwise its hard to interpret the description.
- Could you include the angle of curvature of the teeth as this is often included in many cryptoclidid tooth descriptions
- Is there any evidence of wear facets?
- Please include full names of taxa (genus and species name)

---

## Round 0.2 · Major Revisions

Dear authors,

Both reviewers have made a decision of 'major revisions', which I have accepted.

Both reviewers still have concerns about the referral to Cryptoclididae, and some details on your ordination analyses.

I look forward to receiving your revised manuscript.

·

Basic reporting

Thank you for the opportunity to review the revised version of the manuscript. I am happy to see that the authors modified the MS thoroughly and applied analytics to assess the morphospace occupation of the studied material among Jurassic marine reptiles. For my comments regarding the methods and interpretations see below.

In general, I certainly find the MS appropriate for PeerJ and genuinly want to see it published. Nevertheless, I've still got the impression that the MS is at least three times as long as necessary. Numerous parts are speculative (especially with respect to the taxonomic assignment of the specimens and their paleobiogeographic significance) or do not add to the study much (there is a lot of "comments" on plesiosaurs and "explanations" of the methods that do not seem to be relevant and are often wrong or too simplified). See the attached PDF with my comments to particular sections.

Experimental design

The authors used a principal coordinates analysis (PCoA) with the data of Foffa et al. (2018), and considered its results as support for assigning the material to Cryptoclididae. This method, however, did not provide such support. Their results only show that the material occupies the plesiosauroid dental morphospace (which includes cryptoclidids). Plesiosauroids are generally poorly represented in the dataset and even though I still agree that the placement within Cryptoclididae is the most likely (because of the morphology and the age of the material), the authors went way too far for example by considering the material closely related to Tricleidus. By the way, if you want to look at similarities (that do not necessarily mean relationships), you may use the same data to run a cluster analysis. Still, even if that analysis places your material closely to Cryptoclididae or within its few representatives that are included in the dataset, you probably shouldn't go further than with Cryptoclididae? indet.

Validity of the findings

All underlying data have been provided. See above and the attached PDF for comments on the findings.

Additional comments

My biggest problem with the MS is that huge amounts of text (especially the "facts" and "explanations") could be removed from the text without actually loosing the significance. The study reports interesting specimens that I strongly believe should be published (and hope to see published). But the MS would be much improved if the text was more concise; just stating what you've got, what methods you used, providing descriptions and comparisons, providing results of the PCoA (and cluster analysis, if you want to assess similarities to particular taxa in the dataset), and that's it. There is no point to discuss paleobiogeography because your study does not add to the discussion without larger-scale analyses of all faunal components forming different European assemblages.

To sum it up, I definitely recommend acceptance of the MS but I think that major revisions are still needed to make the MS acceptable for publication.

Reviewer 2 ·

Basic reporting

The incorporated changes from the first draft are good, but there is still quite a bit of sentence restructuring and grammar correcting that needs to be done. I have highlighted much of the manuscript where changes need to be made, but didn't make it through the entire manuscript. The article is structured in an acceptable manner with raw data shared. I am having a difficult time however accepting the hypothesis that the teeth can be ascribed to Cryptoclididae. I have a question about the raw data; in the supplementary file, I count 12 plesiosauroid taxa (highlighted in light blue) but in Fig. 3A, I see 13 plesiosauroid taxa occupying the morphospace.

Experimental design

It is good that there was much more detailed comparative work in this manuscript and including the PCoA adds for a quantitative comparison. Good description of methods used. I would have liked to see a broader comparative section, comparing the new teeth with cryptoclidid taxa outside of Europe, maybe you can find a morphological match comparing the teeth to cryptoclidid species in other parts of the world? How do we know the teeth aren't rhomaleosaurid? How about microcleididae? I'm just suggesting to keep an open mind.

Validity of the findings

My biggest critique is that I am not convinced the newly described teeth can be ascribed to Cryptoclididae based on fact that the teeth look distinct from all cryptoclidid teeth in the comparison section. It is argued in the text that the newly described teeth probably belong to Tricleidus but I don't see how. As stated in the text, the curvature and compression of the new teeth are distinct from Tricleidus- it is argued that compressional deformed the teeth to produce the labiolingual compression. I sincerely doubt this (although of course not impossible), as all four teeth are compressed labiolingually, thus the forces would have to have acted identically on all four teeth. The argument for Tricleidus stems from the results of the PCoA, which also presents some issues (in my opinion). In Fig. 3A, 11/OB/T4 plots outside of plesiosauroidea, which includes cryptoclidid taxa (such as Tricleidus if I understand correctly). So, the tooth specimen with all the marine reptiles doesn't fall into a clear group. Then, in the second PCoA it is not only falling specifically within Plesiosauroidea, but it is nearly identical to Tricleidus. This jump is interesting, as it shows that given an specific dataset, you can get either inconclusive or more conclusive results (I'm not terribly familiar with using PCoA). This suggests to me that PCoA can in fact be misleading to some degree. Another jump is made, where Fig. 3B suggests 11/OB/T4 is essentially identical to Tricleidus in the morphospace, but then looking at the gross morphology, the teeth are clearly distinct. As one last jump, compression is cited for the reason the teeth look different. These jumps in interpretation are problematic. I think best case, from what I have seen presented thus far, the new teeth are plesiosauroid indet. with a very tentative suggestion for a cryptoclidid affinity. Even in the best hypothetical case, let's say 11/OB/T4 is morphologically identical to Tricleidus (i.e., there's no compression or curvature). Even in that case, there is still room for skepticism, because as you admit in your text identifying taxa from a single tooth is challenging and you have problems with homoplasy. Plesiosaurians are a diverse group of marine reptiles and they are prolific in the Jurassic. You can have diverse taxa display similar morphology.

Additional comments

Thank you very much for the revised manuscript. Please see the comments for how to improve the writing.

Annotated reviews are not available for download in order to protect the identity of reviewers who chose to remain anonymous.

---

## Round 0.3 · Minor Revisions

Dear authors,

Both reviewers have been very complimentary about your revised manuscript. Reviewer two has made some grammatical changes to your manuscript that will help its readability.

I look forward to receiving your revised manuscript.

·

Basic reporting

I would like to thank the authors for their detailed response to all my comments. I am happy to see that the authors adopted a more cautious interpretation of the taxonomic affinities of the material and the results of the PCoA. Despite that there are still parts that I think are not strictly necessary, do not add to the merit of the article, or are too speculative, I don't want to delay the publication of this study. I look forward to seeing it out.

Experimental design

/

Validity of the findings

/

Additional comments

/

Reviewer 2 ·

Basic reporting

Thank you for an improved manuscript. I have highlighted areas that need rewriting, rephrasing, and corrections for grammar. I recommend going through the manuscript with a 'flea comb' to make sure all the grammar is correct.

Experimental design

Given the four isolated teeth, the description with comparisons and PCoA analysis are the best ways to study the teeth and give a possible taxon ID. The tentative suggestion for Cryptoclididae is acceptable.

Validity of the findings

Your conclusions are fine, however, in the abstract there is a statement about how the new site reflects Tethyan and Boreal taxa mixing based on the teeth. However, we cannot say this as the teeth do not have a clear identification.

Additional comments

Excellent job with the revisions. Just check the grammar, address the mixing statement in the abstract, and clarify some thoughts and statements in the manuscript. Once that is cleared, my recommendation is to publish this work.

Annotated reviews are not available for download in order to protect the identity of reviewers who chose to remain anonymous.

---

## Round 0.4 · accepted · Accept

Dear authors,

Thank you for your revised manuscript. Based on your rebuttal letter, and how you addressed reviewer comments I am happy to accept your manuscript.

The production staff will contact you to take you through the proofing stage.

Congratulations, and I hope you will choose PeerJ as your publication venue again in the future.